# Pattern Recognition of Varieties of Peach Fruit and Pulp from Their Volatile Components and Metabolic Profile Using HS-SPME-GC/MS Combined with Multivariable Statistical Analysis

**DOI:** 10.3390/plants11233219

**Published:** 2022-11-24

**Authors:** Dasha Mihaylova, Aneta Popova, Ivayla Dincheva

**Affiliations:** 1Department of Biotechnology, Technological Faculty, University of Food Technologies, 4002 Plovdiv, Bulgaria; 2Department of Catering and Nutrition, Economics Faculty, University of Food Technologies, 4002 Plovdiv, Bulgaria; 3Department of Agrobiotechnologies, AgroBioInstitute, Agricultural Academy, 1164 Sofia, Bulgaria

**Keywords:** peach fruit and pulp, volatile composition, HS-SPME-GC-MS, metabolic chemotaxonomy

## Abstract

A fruit’s aroma profile, composed of a complex mixture of volatile organic compounds, is among the core attributes related to the overall taste and consumer preference. *Prunus persica* L. is a preferred summer fruit with a distinct, favorable olfactory characteristic. The volatile compositions of both peach fruits and fruit pulps from eight peach cultivars (four native and four introduced) was investigated to compare their composition and assess flavor-contributing compounds. In total, 65 compounds were profiled after a HS-SPME-GC-MS analysis: 16 esters, 14 aldehydes, 5 alcohols, 7 hydrocarbons, 7 ketones, 8 acids, and 8 terpenes. The most common compounds were esters, acids, and aldehydes. Although the same compounds were identified in both fruit and pulp, their %TIC (total ion current) differed in favor of the whole fruit. Following the metabolic profiling of the whole fruit and fruit pulp, a total of 44 compounds were identified from the studied varieties. Among them, amino acids, organic acids, sugar alcohols, saccharides, fatty acids, and phenolic acids were identified as existing groups. According to the provided principal component analysis (PCA) and hierarchical cluster analysis (HCA), the relative %TIC of the identified volatile compounds fluctuated depending on the studied cultivar. No differences were visible in the PCA biplots, which suggested that the polar and lipid metabolites do not provide significant variations when considering different parts of the fruit, contrary to the volatile compounds. The obtained results could successfully be applied in the metabolic chemotaxonomy of peaches and the differentiation of the metabolites present in different parts of the peach.

## 1. Introduction

Fruit consumption is very important for general health and wellbeing, with a number of national strategies focusing on its increased intake. Although components like sugar content, element composition, fibers, phenolic acids, and organic acids will contribute positively to the health of the average individual, volatile compounds are the ones that will provoke the desire to eat. Volatile compounds are extremely important when the consumer makes the final decision about foods that will be eaten. The peach fruit is one of the most preferred fruits worldwide, consumed both fresh and processed. The volatile profile of peaches has been studied with various aims throughout the years [1,2,3].

Different methods are applied in volatolomics. Currently, the HS-SPME-GC-MS has been recognized as a useful method that drastically minimizes interference during identification [4]. Prior to analysis, the extraction process has to be chosen carefully in order to result in a proper determination [5]. The taste/aroma qualities of fruits, and peaches in particular, are highly dependent on the volatile and semi-volatile organic compounds present in both the sample matrix and the headspace aroma. Approximately 100 volatiles have been identified in the peach fruit [6]. These chemicals can be grouped into the following groups based on their constituent structures: aldehydes, alcohols, terpenoids, esters, and lactones. Volatiles are mainly derived from precursors such as fatty acids, terpenes, and amino acids [7]. Peach fruit volatile compounds are highly dependent on genetic origin. For example, a study reported a wide variation of volatiles among 50 peach cultivars [1]. In addition, peach fruits with different flesh color are described as varying in their volatile profiles at ripe stages. Red-fleshed peach fruits accumulate different volatile concentrations compared to white-fleshed ones [8]. However, these differences have not led to the distinguishment of cultivars based on their fruit pulp or whole fruit. Most of the studies in the literature are still focused on peaches from different cultivars. Recent papers are beginning to valorize the peel with its beneficial properties [9], but the peach pulp is not an object of extensive research interest and thus creates an interesting niche for investigation [10].

Metabolic profiling is often used for plant characterization [11,12]. Studies focus on both climacteric and non-climacteric fruits, as well as vegetables [13,14]. To date, few studies have divided samples to their individual components, i.e., skin, pulp, and stone, among others [15,16]. Plants comprise of a wide variety of compounds, which makes a single analytical approach insufficient in terms of simultaneous extraction of all metabolites present in a sample [17]. Metabolic variability proves that many conditions, i.e., geographical region, climate peculiarities, variety, soil specificity, etc., can be considered important when investigating targeted and untargeted metabolites [18].

Peaches (*Prunus persica* (L.) Batch.) are grown in different regions throughout Bulgaria, and in 2020, the relative share of the total production of peaches and nectarines was 10.2%. More precisely, 20,740 tons of peaches were produced in 2020, which was a drop of −38.8% compared to 2019 [19], which can be attributed to the COVID-19 crisis and the restrictions introduced in relation to it.

A wide range of culinary applications appeared over the last few years, which indicates the increasing interest toward this fruit. Peaches are eaten fresh and used as ingredients in value-added, processed products [20,21]. The consumer’s interest is based not only on the personal perception of the fruit, but also on the contemporary striving for a healthy lifestyle. This is due to the more and more information regarding fruit composition advantages. The phytocompounds in fruits, and peaches in particular, are associated with several health benefits—antidiabetic effects, anti-obesity effects, hypertension control, anti-inflammatory effects, antioxidant activity, prevention of neurogenerative diseases (Alzheimer’s), etc. [22,23,24].

The aim of the current study was to investigate the volatile compositions and metabolic profiles of both peach fruit and fruit pulp from eight peach cultivars, and to provide comparable data relevant to the existing knowledge on the subject of peach volatolomics and metabolomics. With the aid of principal component analysis (PCA) and hierarchical cluster analysis (HCA), differentiation and clustering of the results was provided in terms of the metabolites present in different parts of the peach. This work may represent an interesting approach for future studies in line with trending topics like zero-waste management.

## 2. Results and Discussion

The metabolic composition of peaches has been previously reported [25,26], but none of the widely available research has made a differentiation between the compounds present in the whole fruit and those typical of the fruit pulp only. Thus, the current study follows such an approach for characterization. A total of 44 compounds were identified from the whole fruit and fruit pulp from the studied varieties (Table 1). Among them, amino acids, organic acids, sugar alcohols, saccharides, fatty acids, and phenolic acids were identified as existing groups. Sometimes, the use of diverse experimental approaches leaves researchers challenged when comparison of data should be applied.

Organic acids are generally important to the specific tastes of fruits due to the sourness and acidity they provide [27]. The eight studied varieties ranged in terms of shikimic acid, malic acid, citric acid, quinic acid, and succinic acid, with the WF having higher levels compared to the FP. The primal organic acid in all samples (FP and WF) was shikimic acid, while the least present was L-ascorbic acid. Shikimic acid has been previously reported as most present in peach varieties [28]. Similarly, malic acid, citric acid, quinic acid, and succinic acid are regularly reported as the dominant ones [29,30,31].

Phenolic compounds are considered to be widely distributed in plants [32]. The current results (Table 1) reveal the presence of six phenolic acids, while only one of the varieties (“July Lady”) lacked p-coumaric acid. Studies acknowledge the importance of phenolic acids due to their biological functions [33]. Here, once more the whole fruit sample is found to be richer in phenolic acids compared to the fruit pulp. This proves that the fruit skin should be an object of research and consumed where appropriate due to its ability to accumulate health-contributing compounds. Protocatechuic acid, chlorogenic acid, and caffeic acid are most present in the WF and FP of the “Gergana” variety, p-coumaric acid and sinapic acid in the “Filina”, and ferulic acid in the “Flat Queen”. By default, chlorogenic acid is reported as the most abundant in plant samples [34], which is further supported by the current results. Chlorogenic acid is reported as a possible preventive of cardiovascular disease [35].

The amount of amino acids established in the current study reveals that peaches cannot be considered protein contributors in the daily diet. A total of 18 amino acids have been identified in the peach varieties object of analysis. From all studied samples, the two nectarine varieties (“Gergana” and “Morsiani 90”) are the only ones that contain all essential amino acids in even limited quantities. Aspartic acid is the least detected in all samples. The same trend as for other compounds, where the WF is richer than the FP, is observed for the amino acids.

In accordance with earlier studies [36], linoleic acid is reported as one of the most abundant fatty acids. Other fatty acids (FA) found in greater quantities are the palmitic, oleic, and stearic acids. The WF samples are documented as richer in FAs compared to the FP ones. The “Laskava”, “July Lady”, and “Morsiani 90” varieties hold the most FAs compared to the other studied varieties. There is a predominance of long-chain saturated fatty acids and the presence of more PUFA compared to MUFA. Based on existing reports [37,38], these findings hint at the possibility of more profound studies on the topic of the possible health effects of the fatty acid content in peaches.

The investigated volatile compositions of both peach fruit and fruit pulp from eight peach cultivars are presented in Table 2. In total, 65 compounds were profiled after an HS-SPME-GC-MS analysis: 16 esters, 14 aldehydes, 5 alcohols, 7 hydrocarbons, 7 ketones, 8 acids, and 8 terpenes. The highest content of volatile compounds was attributed to the “Morsiani 90” variety (Table 2) (both pulp and whole fruit), while the lowest belonged to the “Ufo 4” (whole fruit) and “July Lady” (fruit pulp). More volatiles were generally present in the whole fruit of peaches, contrary to flat peaches and nectarines, where more volatiles were identified in the fruit pulp. None of the metabolites was identified in only either pulp or whole fruit. However, the whole fruit was richer in terms of %TIC of the identified compounds, which undoubtedly points to the importance of the peel for the flavor experience and is a favorable argument for consuming the fruit without peeling it. The chemical families that were more abundant in the fruit pulp were hydrocarbons, alcohols, and ketones.

The compositions and concentrations of volatile compounds and metabolites in the peach fruits and pulps are shown in Figure 1. The profile shows that the samples consist of the same chemical classes. However, total emission is not only variety-dependent but also fluctuates depending on the presence or absence of the fruit skin.

Some of the identified volatiles in the studied peach varieties belong to the so-called microbial volatiles [39], which are natural substances that enhance plant growth, productivity, and disease resistance. A few of them are butanoic acid (high % of TIC in the “Laskava” and “July Lady” varieties), benzaldehyde (relatively low in all studied samples), hexanol (relatively low in all studied samples), and myrcene.

Saccharides and fatty acids are the most frequent compounds in the WF and FP of the studied peaches, with different percentage distributions (Figure 1). The least found compounds are the sugar alcohols. Aldehydes and esters were the most present compounds in the whole fruits of the studied varieties (Figure 1). Alcohols, on the other hand, were the least found in all samples. Previous research on the WFs of peach varieties has established a similar trend in the results [2]. Analogous compound groups have also been identified in a recent study of famous Chinese peach cultivars [40].

Considering their abundance, the most important aldehydes for the WF and FP are (E)-2-hexenal, hexanal, heptanal, and nonanal. Contrary to the current results, benzaldehyde was identified as a major aldehyde contributor in a study of American peach cultivars [3]. Heptanal has its highest quantity in the early-ripening variety “Filina”. Hexanal and (E)-2-hexenal are compounds which are associated with the maturation process, and their quantities are lower in the FPs of all studied varieties. Hexanal was found to be naturally occurring within the fruit and is considered important for the volatile profile of nectarines, in particular [41]. Its %TIC in the two studied nectarine varieties is rather similar, even though the ripening period is different. Both hexanal and E-2-hexenal are also regularly identified in apple cultivars and are associated with a specific leafy–sweet odor [42].

According to other research, the acid content in peaches is rather low [43]. The most dominant acids in the investigated peach whole fruits and pulps are dodecanoic acid and nonanoic acid. Ketones come in relatively low amounts, with γ-octalactone and γ-dodecalactone being the most abundant. Lactones are usually isolated in peach volatiles, and they are flavor-contributing [26]. Lactones are also linked to a specific peach-like odor and flavor [44]. According to the established results, the “Ufo 4” and “July Lady” should exhibit the most distinct peach-associated aroma. It should also be mentioned that the fruit pulp contains more lactones compared to the whole fruit. Γ-Dodecalactone is also a major lactone isolated in other research [3]. Lactones, particularly delta-lactones, are as well implicated in the peach aroma. Recently, benzaldehyde, linalool, and C_10_ lactones were found to increase in the final period of peach ripening, while C_6_ aldehydes decreased [2,45].

A total of five alcohols were Identified in all studied samples. The relative cumulative alcohol content ranged from 2.69 %TIC (“Filina”, WF) to 6.73 %TIC (“Evmolpiya”, FP). The highest alcohol content is found in the “Evmolpiya” variety, in both the WF and FP. The content of benzyl alcohol, which is a natural component in peaches [46], gradually increases for the varieties that ripen in the months of July and August and then decreases in the late-ripening “Flat Queen” and “Evmolpiya” varieties. The “Morsiani 90” does not fall into this trend, as it is the latest ripening (mid-September) variety used in this study, and a steady increase in its benzyl alcohol content is registered. This may suggest that not only the ripening period, but the cultivar as well contributes to the volatile peculiarities of each variety.

Esters contributed largely to the overall volatile components identified in this study. Moreover, the most abundant esters (average content >3.00%TIC) as determined by GC-MS were ethyl tiglate, ethyl hexanoate, ethyl benzoate, and 2-phenyl ethyl butanoate. Methyl decanoate and benzyl butanoate were present in relatively low amounts (average content <1.5% TIC) in the FPs of each variety. Consistent with previous reports [47,48], ethyl tiglate was the most abundant in most of the WFs of the cultivars analyzed in this study.

The estimated amount of terpenes identified in the studied peach varieties is less than that estimated in the peach cultivar “Cresthaven” grafted onto five different rootstocks [49]. However, a resemblance in the compounds found in the highest quantities (limonene, linalool, and ocimene) is visible. This indicates that the cultivar, seasonal dynamics, and growing conditions influence the volatiles emitted by the same fruit, but there are some specific compounds that can be found in all cultivars.

Among the seven identified hydrocarbons, the amount of tetradecane stood out from the others. It was also observed that the nectarine samples were richer in tridecane compared to flat peaches and peaches. These results comply with and extend the ones published about peach volatile emissions from Chinese samples [50].

Aroma is very valuable for the consumer, and specific attention should be paid to odor-important compounds. Thus, if key odor-active compounds for peaches have to be identified, lactones will be among them [51]. Although past research [52] on the relative VOC concentrations and sensory attributes of peaches found higher concentrations of monoterpenes and esters compared to lactones in peaches, the latter are major contributors to the specific fruity, peach-like aroma [53]. Terpenoids and aldehydes are seen as insignificant contributors [53]. The sensory quality of peaches and fruit in general is also dependent on the odor threshold [54]. This may be the reason for odor-active substances’ inability to produce flavor intensity among the overall volatile compounds.

With reference to the abovementioned knowledge, fruit skin is identified as an extremely important contributor to the overall sensory experience in terms of lactone content. Consumption of peaches with their skin will not only contribute to the zero-waste approach promoted nowadays but will also maximize the sensory experience of the consumer.

### Principal Component Analysis (PCA) and Hierarchical Cluster Analysis (HCA) of HS-SPME-GC-MS Data

PCA is a commonly used dimensionality-reduction method that provides visualizations of data. As is evident from Figure 2, the studied cultivars were divided into different groups when the WF or only the FP was being evaluated. This undoubtedly proves the importance of the fruit skin to the overall volatile contribution.

The applied multivariable analysis aided in the identification of the volatiles that contribute to the differentiation of the peach varieties’ WFs and FPs. High positive load scores in PC1, which distinguish the “Flat Queen” from the other studied peaches, are shown by dodecane, nonanal, and ethyl octanoate. The high negative scores in PC1 clearly differentiate the “July Lady” and “Laskava” whole fruits from the others. When the fruit pulp is assessed, high positive load scores in PC1 distinguish again the “Flat Queen” from the other studied varieties in terms of 1-octen-3-yl-butanoate. The “Ufo 4” pulp appeared different from the rest due to the negative scores of several compounds in PC2.

PC1 and PC2 explain 44.4% of the total variation in the peaches’ whole fruits, while the differences in the pulps when considering PC1 and PC2 account for 58.1% of the variation. The clustering results also confirm the differences when taking into account the WF or only the FP. The conducted analysis of the data revealed that peaches, flat peaches, and nectarines can fall into the same cluster when the contribution of their volatiles is being assessed.

Considering the clade arrangement in Figure 2A,B, it can be concluded that when the fruit pulp is being evaluated, more differences are visible between cultivars compared to evaluation of the whole fruit. In both dendrograms, there are seven clades consisting of different arrangements when evaluating the WF and FP. For example, the fruit pulp leaves the “Morsiani 90” and “July Lady” as clustered together, being the most different from the others. On the other hand, the whole fruit reveals the “Filina” and “Flat Queen” as the most different from the others, but similar to each other. Here, once more, the importance of the skin is observed in the differences of the data visualization.

The same approach was used in terms of the polar and lipid metabolites, and the results are presented in Figure 3. PC1 and PC2 explain 63.4% of the total variation in the peaches’ whole fruits and FPs. No differences are visible in the PCA biplots, which suggests that the polar and lipid metabolites do not provide significant variations when considering different parts of the fruit, contrary to the volatile compounds. Saccharides, being the most frequent compounds in the WFs and FPs of the studied peaches, exhibit high positive load scores.

Considering the clade arrangement, it can be assumed that when the fruit pulp is being evaluated, more differences are visible between cultivars compared to evaluation of the whole fruit. In both dendrograms (Figure 3A,B) there are six clades consisting of the same arrangements when evaluating WFs and FPs. The “July Lady” is simplicifolious, which is significantly different from the arrangement when volatiles are being evaluated. The “Flat Queen” and “Morsiani 90” are revealed as the most different from the others, but they are similar to each other.

## 3. Materials and Methods

### 3.1. Fruit Samples

Both native and introduced varieties of peaches (“Filina”, “Laskava”, “July Lady”, and “Evmolpiya”), flat peaches (“Ufo 4” and “Flat Queen”), and nectarines (“Gergana” and “Morsiani 90”) were used in the current study. Ripe fruits were collected in the 2021 season at the Fruit Growing Institute, Plovdiv, Bulgaria. Relevant samples (whole fruit and fruit pulp only) were first freeze-dried with a vacuum freeze dryer (BK-FD12S, Biobase, Shandong, China); after that, they were powdered and kept prior to extraction.

### 3.2. GC-MS Headspace-Solid Phase Micro Extraction (HS-SPME)

Gas chromatography–mass spectrometry (GC-MS) was used to investigate the two fractions obtained—the polar (amino and organic acids, carbohydrates) and non-polar (saturated and unsaturated fatty acids) fractions.

Briefly, 50.0 mg lyophilized peach material from each sample was subjected to the procedure described by Iantcheva et al. [55]. The GC-MS analysis was carried out using a 7890A gas chromatograph (Agilent, Santa Clara, CA, USA) coupled to a 5975C mass-selective detector (Agilent) and an HP-5ms silica-fused capillary column coated with a 0.25 µm film of poly(dimethylsiloxane) as the stationary phase (Agilent), with dimensions of 30 m × 0.25 mm (i.d.). The oven temperature program used was as follows: initial temperature 100 °C for 2 min, then 15 °C/min to 180 °C for 2 min, then 5 °C/min to 300 °C for 10 min, with a run time of 42 min. The flow rate of the carrier gas (helium) was maintained at 1.2 mL/min. The injector and the transfer line temperature were kept at 250 °C, with EI energy at 70 eV and a mass range of 50 to 550 m/z at 1.0 s/decade. The temperature of the MS source was 230 °C. The injections were carried out in a split mode at a 10:1 ratio; the injection volume was 1 µL.

For the purposes of headspace sampling, 2 cm of SPME fiber assembly divinylben-zene/carboxen/polydimethylsiloxane (DVB/CAR/PDMS, Supelco, Bellefonte, PA, USA) was used. The HS-SPME technique was used for the analyses of the peach volatile compounds according to the procedure described by Uekane et al. [56].

An Agilent 7890A GC unit coupled to an Agilent 5975C MSD and a DB-5ms (30 m × 0.25 mm × 0.25 μm) column were used to analyze the volatile compounds in all investigated samples. The oven temperature program was as follows: from 40 °C (hold 1 min) to 250 °C (hold 5 min) at 2 °C /min; carrier gas: helium with a flow rate of 1.2 mL/min; transfer line temperature: 270 °C; ion source temperature: 200 °C, EI energy: 70 eV; and mass range: 50 to 550 m/z at 1.0 s/decade.

The AMDIS software, version 2.64 (Automated Mass Spectral Deconvolution and Identification System, NIST, Gaithersburg, MD, USA), aided in the reading of the obtained mass spectra and the identification of the metabolites. AMDIS recorded the RIs of the compounds with a standard n-hydrocarbon calibration mixture (C8–C36, Restek, Teknokroma, Barcelona, Spain). For identification, the separated compounds were compared to their GC-MS spectra and Kovats retention indexes (RI) using reference compounds in the NIST’08 database (NIST Mass Spectral Database, PC-Version 5.0, 2008 from National Institute of Standards and Technology, Gaithersburg, MD, USA).

### 3.3. Statistical Analysis

PCA and HCA of GC-MS data were conducted using MetaboAnalyst—a web-based platform (www.metaboanalyst.ca, accessed on 27 June 2022)—as described by Mihaylova et al. [2].

## 4. Conclusions

A fruit’s aroma profile, composed of a complex mixture of volatile organic compounds, is among the core attributes related to the overall taste and consumer preference. This is probably one of the key reasons for the scientific interest in the field of volatolomics. *Prunus persica* L. is a preferred summer fruit with a distinct, favorable olfactory characteristic. The volatile compositions of both peach fruits and fruit pulps from eight peach cultivars (four native and four introduced) were investigated to compare their compositions and assess flavor-contributing compounds.

Overall, 65 compounds were profiled after an HS-SPME-GC-MS analysis: 16 esters, 14 aldehydes, 5 alcohols, 7 hydrocarbons, 7 ketones, 8 acids, and 8 terpenes. More volatiles were generally present in the whole fruits of peaches, contrary to flat peaches and nectarines, in which more volatiles were identified in the fruit pulps. None of the metabolites was identified in only either pulp or whole fruit. However, the whole fruits were richer in terms of the quantities of the identified compounds, which undoubtedly highlights the importance of the peel for the flavor experience. The chemical families that were more abundant in the fruit pulps were hydrocarbons, alcohols, and ketones.

A total of 44 compounds were identified from the whole fruits and fruit pulps of the studied varieties. Among them, amino acids, organic acids, sugar alcohols, saccharides, fatty acids, and phenolic acids were identified as existing groups. From the metabolites, the amino acid group fluctuated in the samples, with representatives (i.e., aspartic acid, histidine, and tryptophane) missing in several varieties of both WF and FP.

No differences were visible in the PCA biplots, which suggested that the polar and lipid metabolites do not exhibit significant variations when considering different parts of the fruit, contrary to the volatile compounds. The applied PCA divided the studied cultivars into different groups when volatiles of the WF or only the FP were being evaluated. This undoubtedly proves the importance of the fruit skin to the overall volatile contribution. Fruit skin was identified as extremely important to the overall sensory experience in terms of lactone content. Consumption of peaches with their skin will not only contribute to the zero-waste approach promoted nowadays but will also maximize the sensory experience of the consumer. The obtained results could successfully be applied in the metabolic chemotaxonomy of peaches and differentiation of the metabolites present in different parts of the peach.

## Figures and Tables

**Figure 1 plants-11-03219-f001:**
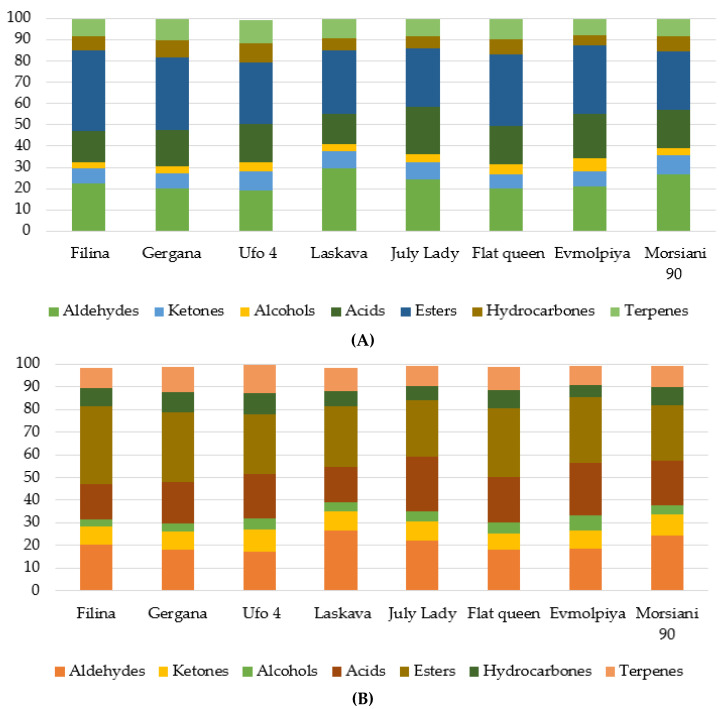
Distribution of volatile compounds ((**A**)—WF, (**B**)—FP) and metabolites ((**C**)—WF, (**D**)—FP) according to their chemical families in eight peach cultivars.

**Figure 2 plants-11-03219-f002:**
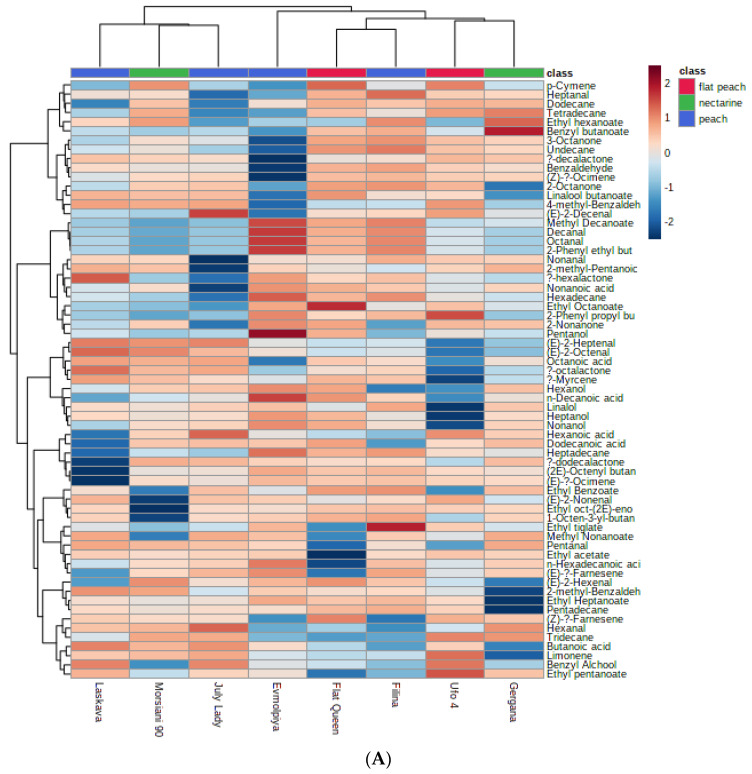
Clustering results for eight peach cultivars, shown as a heatmap: (**A**)—WF, (**B**)—FP. The color scale of the heatmap ranged from dark brown (value, +2) to dark blue (value, −2). The values were normalized by log_10_ transformation.

**Figure 3 plants-11-03219-f003:**
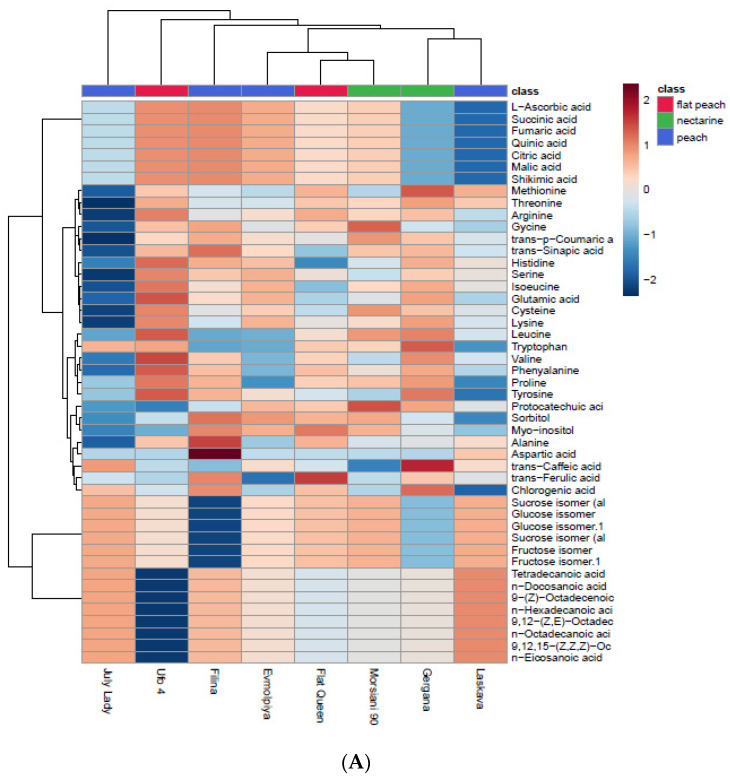
Clustering results for eight peach cultivars, shown as a heatmap: (**A**)—WF, (**B**)—FP. The values were normalized by log_10_ transformation.

**Table 1 plants-11-03219-t001:** Metabolites identified in peach fruit (WF) and pulp (FP) of 8 varieties analyzed by HS-SPME-GC-MS. The results are given in mg/g DW.

RI	Name	Filina		Gergana		Ufo 4		Laskava		July Lady		Flat Queen		Evmolpiya		Morsiani 90	
		WF	FP	WF	FP	WF	FP	WF	FP	WF	FP	WF	FP	WF	FP	WF	FP
**Amino acids**																	
1097	Alanine	1.24 ± 0.09	0.98 ± 0.12	0.18 ± 0.01	0.14 ± 0.01	0.37 ± 0.03	0.29 ± 0.03	0.30 ± 0.02	0.24 ± 0.02	0.02 ± 0.00	0.02 ± 0.00	0.40 ± 0.03	0.32 ± 0.10	0.07 ± 0.03	0.06 ± 0.01	0.15 ± 0.02	0.12 ± 0.01
1208	Valine	0.25 ± 0.05	0.20 ± 0.05	0.54 ± 0.02	0.43 ± 0.03	0.98 ± 0.11	0.76 ± 0.04	0.14 ± 0.02	0.11 ± 0.01	0.02 ± 0.00	0.02 ± 0.00	0.22 ± 0.02	0.17 ± 0.02	0.05 ± 0.01	0.04 ± 0.01	0.09 ± 0.01	0.07 ± 0.01
1266	Leucine	nd	nd	0.09 ± 0.00	0.07 ± 0.01	0.14 ± 0.08	0.11 ± 0.02	0.01 ± 0.00	0.01 ± 0.00	nd	nd	0.02 ± 0.00	0.01 ± 0.00	nd	nd	0.06 ± 0.01	0.05 ± 0.01
1285	Isoeucine	0.21 ± 0.05	0.16 ± 0.02	0.47 ± 0.01	0.37 ± 0.05	0.77 ± 0.08	0.60 ± 0.02	0.21 ± 0.01	0.16 ± 0.01	0.02 ± 0.00	0.01 ± 0.00	0.06 ± 0.01	0.05 ± 0.01	0.37 ± 0.03	0.29 ± 0.03	0.25 ± 0.02	0.20 ± 0.03
1293	Proline	0.28 ± 0.01	0.22 ± 0.13	0.52 ± 0.01	0.41 ± 0.05	0.92 ± 0.07	0.73 ± 0.02	nd	nd	0.01 ± 0.00	0.01 ± 0.00	0.14 ± 0.06	0.11 ± 0.02	nd	nd	0.20 ± 0.01	0.16 ± 0.04
1299	Gycine	0.39 ± 0.04	0.31 ± 0.04	0.14 ± 0.01	0.11 ± 0.03	0.31 ± 0.05	0.25 ± 0.01	0.10 ± 0.01	0.08 ± 0.01	nd	nd	0.24 ± 0.04	0.19 ± 0.02	0.14 ± 0.01	0.11 ± 0.02	0.69 ± 0.04	0.54 ± 0.03
1351	Serine	1.04 ± 0.07	0.82 ± 0.06	1.09 ± 0.05	0.86 ± 0.05	2.18 ± 0.05	1.72 ± 0.09	0.77 ± 0.05	0.61 ± 0.05	0.05 ± 0.01	0.04 ± 0.01	0.63 ± 0.06	0.50 ± 0.03	1.25 ± 0.19	0.99 ± 0.05	0.42 ± 0.03	0.33 ± 0.02
1376	Threonine	0.35 ± 0.03	0.28 ± 0.03	1.64 ± 0.06	1.29 ± 0.06	1.37 ± 0.05	1.08 ± 0.05	1.06 ± 0.07	0.83 ± 0.05	0.02 ± 0.01	0.02 ± 0.00	0.84 ± 0.08	0.66 ± 0.02	0.31 ± 0.04	0.25 ± 0.03	0.73 ± 0.04	0.58 ± 0.05
1508	Aspartic acid	2.01 ± 0.25	1.59 ± 0.16	nd	nd	nd	nd	0.12 ± 0.03	0.09 ± 0.01	nd	nd	nd	nd	nd	nd	nd	nd
1515	Methionine	0.09 ± 0.01	0.07 ± 0.01	0.71 ± 0.03	0.56 ± 0.04	0.22 ± 0.02	0.17 ± 0.03	0.34 ± 0.02	0.26 ± 0.02	0.01 ± 0.00	0.01 ± 0.00	0.25 ± 0.01	0.20 ± 0.01	0.06 ± 0.01	0.05 ± 0.01	0.06 ± 0.01	0.05 ± 0.01
1550	Cysteine	0.01 ± 0.00	0.01 ± 0.00	0.02 ± 0.00	0.02 ± 0.00	0.04 ± 0.00	0.03 ± 0.00	0.01 ± 0.00	0.01 ± 0.00	nd	nd	0.01 ± 0.00	0.01 ± 0.00	0.02 ± 0.00	0.01 ± 0.00	0.03 ± 0.01	0.03 ± 0.01
1609	Glutamic acid	0.49 ± 0.03	0.39 ± 0.03	0.92 ± 0.07	0.72 ± 0.03	1.51 ± 0.23	1.20 ± 0.14	0.31 ± 0.02	0.24 ± 0.01	0.09 ± 0.02	0.07 ± 0.02	0.24 ± 0.02	0.19 ± 0.01	0.69 ± 0.03	0.54 ± 0.03	0.38 ± 0.02	0.30 ± 0.03
1635	Phenyalanine	0.38 ± 0.06	0.30 ± 0.02	0.50 ± 0.01	0.39 ± 0.02	0.82 ± 0.05	0.65 ± 0.08	0.19 ± 0.01	0.15 ± 0.01	0.06 ± 0.02	0.05 ± 0.01	0.36 ± 0.02	0.28 ± 0.01	0.10 ± 0.01	0.08 ± 0.02	0.26 ± 0.01	0.21 ± 0.02
1833	Arginine	0.30 ± 0.02	0.24 ± 0.02	0.66 ± 0.02	0.52 ± 0.02	1.25 ± 0.11	0.99 ± 0.08	0.25 ± 0.01	0.20 ± 0.01	0.03 ± 0.00	0.02 ± 0.00	0.73 ± 0.08	0.58 ± 0.03	0.37 ± 0.02	0.29 ± 0.02	0.48 ± 0.02	0.370± 0.04
1910	Lysine	0.32 ± 0.10	0.26 ± 0.02	1.51 ± 0.07	1.19 ± 0.08	1.89 ± 0.13	1.49 ± 0.13	0.43 ± 0.03	0.34 ± 0.03	0.03 ± 0.01	0.02 ± 0.00	0.43 ± 0.08	0.34 ± 0.02	0.99 ± 0.03	0.78 ± 0.05	0.62 ± 0.03	0.49 ± 0.05
1930	Tyrosine	0.27 ± 0.02	0.22 ± 0.06	0.77 ± 0.06	0.61 ± 0.06	1.18 ± 0.12	0.93 ± 0.08	nd	nd	0.02 ± 0.00	0.02 ± 0.00	0.05 ± 0.01	0.04 ± 0.01	0.10 ± 0.01	0.08 ± 0.01	0.03 ± 0.01	0.02 ± 0.00
2144	Histidine	0.68 ± 0.04	0.53 ± 0.02	0.75 ± 0.06	0.59 ± 0.04	1.46 ± 0.10	1.15 ± 0.09	0.36 ± 0.03	0.28 ± 0.03	nd	nd	nd	nd	0.51 ± 0.03	0.40 ± 0.02	0.21 ± 0.02	0.18 ± 0.03
2211	Tryptophan	nd	nd	0.26 ± 0.02	0.21 ± 0.02	0.13 ± 0.06	0.11 ± 0.02	nd	nd	0.12 ± 0.01	0.09 ± 0.01	0.08 ± 0.02	0.06 ± 0.03	nd	nd	0.08 ± 0.01	0.06 ± 0.00
**Organic acids**																	
1305	Succinic acid	1.28 ± 0.05	1.01 ± 0.05	0.96 ± 0.07	0.76 ± 0.05	1.38 ± 0.09	1.09 ± 0.05	0.93 ± 0.05	0.73 ± 0.07	1.09 ± 0.09	0.86 ± 0.06	1.06 ± 0.08	0.84 ± 0.07	1.13 ± 0.06	0.89 ± 0.07	1.17 ± 0.09	0.92 ± 0.06
1344	Fumaric acid	0.96 ± 0.05	0.76 ± 0.05	0.72 ± 0.07	0.57 ± 0.05	1.04 ± 0.04	0.82 ± 0.05	0.70 ± 0.02	0.55 ± 0.05	0.82 ± 0.02	0.65 ± 0.05	0.80 ± 0.06	0.63 ± 0.05	0.85 ± 0.05	0.67 ± 0.06	0.88 ± 0.05	0.69 ± 0.07
1477	Malic acid	1.78 ± 0.06	1.41 ± 0.06	1.34 ± 0.08	1.06 ± 0.12	1.92 ± 0.07	1.51 ± 0.08	1.29 ± 0.11	1.02 ± 0.02	1.51 ± 0.05	1.19 ± 0.06	1.47 ± 0.09	1.16 ± 0.02	1.57 ± 0.07	1.24 ± 0.07	1.62 ± 0.05	1.28 ± 0.07
1818	Shikimic acid	2.93 ± 0.14	2.31 ± 0.22	2.20 ± 0.13	1.73 ± 0.15	3.15 ± 0.21	2.48 ± 0.11	2.11 ± 0.08	1.67 ± 0.08	2.49 ± 0.09	1.96 ± 0.11	2.42 ± 0.11	1.91 ± 0.08	2.58 ± 0.09	2.04 ± 0.12	2.66 ± 0.13	2.10 ± 0.11
1841	Citric acid	1.49 ± 0.11	1.18 ± 0.10	1.12 ± 0.12	0.89 ± 0.01	1.61 ± 0.03	1.27 ± 0.09	1.08 ± 0.08	0.85 ± 0.05	1.27 ± 0.11	1.00 ± 0.10	1.24 ± 0.08	0.97 ± 0.05	1.32 ± 0.06	1.04 ± 0.04	1.36 ± 0.012	1.07 ± 0.05
1855	Quinic acid	1.45 ± 0.10	1.15 ± 0.11	1.09 ± 0.10	0.86 ± 0.05	1.56 ± 0.08	1.23 ± 0.05	1.05 ± 0.03	0.83 ± 0.03	1.23 ± 0.05	0.97 ± 0.02	1.20 ± 0.05	0.95 ± 0.02	1.28 ± 0.05	1.01 ± 0.05	1.32 ± 0.03	1.04 ± 0.06
1946	L-Ascorbic acid	0.50 ± 0.01	0.40 ± 0.05	0.38 ± 0.02	0.30 ± 0.02	0.54 ± 0.02	0.43 ± 0.02	0.36 ± 0.02	0.29 ± 0.01	0.43 ± 0.02	0.34 ± 0.02	0.42 ± 0.02	0.33 ± 0.01	0.44 ± 0.01	0.35 ± 0.02	0.460.03 ±	0.36 ± 0.02
**Sugar alcohols**																	
1932	Sorbitol	0.57 ± 0.01	0.45 ± 0.02	0.43 ± 0.05	0.34 ± 0.05	0.41 ± 0.02	0.32 ± 0.01	0.35 ± 0.03	0.27 ± 0.02	0.32 ± 0.02	0.25 ± 0.03	0.47 ± 0.05	0.37 ± 0.01	0.49 ± 0.05	0.39 ± 0.03	0.52 ± 0.03	0.41 ± 0.02
2034	Myo-inositol	0.33 ± 0.02	0.26 ± 0.02	0.25 ± 0.03	0.20 ± 0.01	0.18 ± 0.01	0.14 ± 0.01	0.22 ± 0.03	0.18 ± 0.02	0.16 ± 0.01	0.13 ± 0.02	0.33 ± 0.05	0.26 ± 0.02	0.28 ± 0.03	0.22 ± 0.03	0.30 ± 0.01	0.24 ± 0.03
**Saccharides (mono-, di-)**																	
1856	Fructose isomer	1.71 ± 0.08	1.35 ± 0.09	2.39 ± 0.09	1.88 ± 0.08	2.80 ± 0.13	2.21 ± 0.23	3.49 ± 0.07	2.78 ± 0.07	3.18 ± 0.09	2.51 ± 0.02	2.62 ± 0.02	2.07 ± 0.04	2.45 ± 0.05	1.93 ± 0.04	2.89 ± 0.02	2.28 ± 0.04
1865	Fructose isomer	0.59 ± 0.04	0.47 ± 0.04	0.83 ± 0.06	0.65 ± 0.04	0.97 ± 0.07	0.76 ± 0.04	1.21 ± 0.04	0.95 ± 0.04	1.10 ± 0.04	0.87 ± 0.02	0.91 ± 0.02	0.72 ± 0.04	0.85 ± 0.02	0.67 ± 0.02	1.00 ± 0.02	0.79 ± 0.02
1881	Glucose isomer	3.27 ± 0.12	2.58 ± 0.09	4.55 ± 0.09	3.59 ± 0.08	5.33 ± 0.14	4.21 ± 0.04	6.67 ± 0.08	5.26 ± 0.04	6.06 ± 0.04	4.78 ± 0.08	5.01 ± 0.05	3.95 ± 0.04	4.67 ± 0.02	3.68 ± 0.01	5.51 ± 0.03	4.35 ± 0.03
1901	Glucose isomer	2.46 ± 0.08	1.94 ± 0.07	3.43 ± 0.12	2.70 ± 0.11	4.02 ± 0.12	3.17 ± 0.03	5.02 ± 0.08	3.96 ± 0.03	4.56 ± 0.03	3.60 ± 0.05	3.77 ± 0.04	2.98 ± 0.03	3.51 ± 0.03	2.77 ± 0.02	4.15 ± 0.03	3.27 ± 0.02
2620	Sucrose isomer (alpha-D-Glc-(1,2)-beta-D-Fru)	5.27 ± 0.09	4.16 ± 0.13	7.34 ± 0.23	5.79 ± 0.13	8.60 ± 0.34	6.79 ± 0.08	10.75 ± 0.34	8.48 ± 0.11	9.77 ± 0.09	7.71 ± 0.08	8.08 ± 0.08	6.37 ± 0.06	7.53 ± 0.07	5.94 ± 0.03	8.89 ± 0.07	7.01 ± 0.05
2833	Sucrose isomer (alpha-D-Glc-(1,2)-beta-D-Fru)	3.06 ± 0.11	2.41 ± 0.11	4.26 ± 0.12	3.36 ± 0.12	4.99 ± 0.23	3.94 ± 0.08	6.24 ± 0.07	4.92 ± 0.09	5.67 ± 0.05	4.48 ± 0.03	4.69 ± 0.05	3.70 ± 0.04	4.37 ± 0.02	3.45 ± 0.02	5.16 ± 0.11	4.07 ± 0.03
**Saturated and unsaturated fatty acids**																	
1719	Tetradecanoic acid (Myristic acid)	0.59 ± 0.03	0.47 ± 0.03	0.56 ± 0.03	0.44 ± 0.03	0.29 ± 0.02	0.23 ± 0.02	0.81 ± 0.03	0.64 ± 0.03	0.69 ± 0.06	0.55 ± 0.04	0.46 ± 0.04	0.36 ± 0.02	0.49 ± 0.03	0.38 ± 0.02	0.51 ± 0.03	0.40 ± 0.01
1926	n-Hexadecanoic acid (Palmitic acid)	6.96 ± 0.04	5.49 ± 0.03	6.61 ± 0.13	5.21 ± 0.05	3.43 ± 0.05	2.71 ± 0.03	9.53 ± 0.15	7.52 ± 0.13	8.14 ± 0.07	6.42 ± 0.12	5.35 ± 0.06	4.22 ± 0.11	5.72 ± 0.14	4.51 ± 0.13	5.95 ± 0.16	4.69 ± 0.22
2095	9,12-(Z,E)-Octadecadienoic acid (Linoleic acid)	4.31 ± 0.04	3.40 ± 0.04	4.10 ± 0.34	3.23 ± 0.05	2.13 ± 0.05	1.68 ± 0.02	5.91 ± 0.11	4.66 ± 0.14	5.05 ± 0.04	3.98 ± 0.10	3.32 ± 0.02	2.62 ± 0.08	3.54 ± 0.11	2.80 ± 0.13	3.69 ± 0.07	2.91 ± 0.05
2099	9-(Z)-Octadecenoic acid (Oleic acid)	2.26 ± 0.03	1.78 ± 0.03	2.15 ± 0.02	1.69 ± 0.03	1.11 ± 0.03	0.88 ± 0.02	3.10 ± 0.08	2.44 ± 0.09	2.65 ± 0.03	2.09 ± 0.04	1.74 ± 0.03	1.37 ± 0.05	1.86 ± 0.2	1.47 ± 0.12	1.93 ± 0.03	1.52 ± 0.05
2103	9,12,15-(Z,Z,Z)-Octadecatrienoic acid (Linolenic acid)	0.78 ± 0.02	0.62 ± 0.01	0.74 ± 0.02	0.59 ± 0.02	0.39 ± 0.02	0.30 ± 0.01	1.07 ± 0.04	0.85 ± 0.04	0.92 ± 0.02	0.72 ± 0.08	0.60 ± 0.03	0.48 ± 0.03	0.64 ± 0.02	0.51 ± 0.05	0.67 ± 0.01	0.53 ± 0.01
2247	n-Octadecanoic acid (Stearic acid)	2.90 ± 0.02	2.29 ± 0.05	2.76 ± 0.02	2.17 ± 0.02	1.43 ± 0.06	1.13 ± 0.0.4	3.97 ± 0.06	3.13 ± 0.03	3.39 ± 0.05	2.68 ± 0.05	2.23 ± 0.05	1.76 ± 0.04	2.38 ± 0.04	1.88 ± 0.05	2.48 ± 0.04	1.96 ± 0.02
2311	n-Eicosanoic acid (Arahydic acid)	1.33 ± 0.02	1.05 ± 0.02	1.26 ± 0.02	1.00 ± 0.02	0.65 ± 0.03	0.52 ± 0.02	1.82 ± 0.04	1.44 ± 0.08	1.56 ± 0.05	1.23 ± 0.03	1.02 ± 0.02	0.81 ± 0.02	1.09 ± 0.02	0.86 ± 0.03	1.14 ± 0.03	0.90 ± 0.02
2408	n-Docosanoic acid (Behenic acid)	1.65 ± 0.02	1.30 ± 0.05	1.57 ± 0.02	1.24 ± 0.03	0.81 ± 0.02	0.64 ± 0.04	2.26 ± 0.04	1.78 ± 0.04	1.93 ± 0.06	1.52 ± 0.05	1.27 ± 0.02	1.00 ± 0.03	1.36 ± 0.04	1.07 ± 0.05	1.41 ± 0.03	1.11 ± 0.03
**Phenolic acids**																	
1836	Protocatechuic acid	0.13 ± 0.01	0.10 ± 0.00	0.33 ± 0.01	0.26 ± 0.01	0.06 ± 0.01	0.05 ± 0.01	0.19 ± 0.01	0.15 ± 0.01	0.07 ± 0.01	0.06 ± 0.01	0.22 ± 0.01	0.17 ± 0.01	0.24 ± 0.01	0.19 ± 0.02	0.48 ± 0.02	0.38 ± 0.01
1945	trans-p-Coumaric acid	0.28 ± 0.01	0.22 ± 0.02	0.25 ± 0.01	0.20 ± 0.01	0.21 ± 0.01	0.17 ± 0.01	0.17 ± 0.01	0.13 ± 0.01	nd	nd	0.15 ± 0.01	0.11 ± 0.01	0.17 ± 0.01	0.13 ± 0.01	0.32 ± 0.01	0.26 ± 0.01
2103	trans-Ferulic acid	0.19 ± 0.01	0.15 ± 0.00	0.16 ± 0.01	0.13 ± 0.01	0.10 ± 0.01	0.08 ± 0.00	0.14 ± 0.01	0.11 ± 0.02	0.11 ± 0.00	0.09 ± 0.01	0.24 ± 0.01	0.19 ± 0.02	0.05 ± 0.00	0.04 ± 0.01	0.10 ± 0.01	0.08 ± 0.00
2140	trans-Caffeic acid	0.06 ± 0.00	0.05 ± 0.00	0.22 ± 0.01	0.18 ± 0.01	0.08 ± 0.01	0.06 ± 0.00	0.12 ± 0.01	0.09 ± 0.01	0.15 ± 0.01	0.12 ± 0.02	0.08 ± 0.00	0.06 ± 0.00	0.09 ± 0.01	0.07 ± 0.00	0.05 ± 0.00	0.04 ± 0.00
2254	trans-Sinapic acid	0.13 ± 0.01	0.10 ± 0.00	0.11 ± 0.01	0.09 ± 0.01	0.11 ± 0.01	0.08 ± 0.00	0.09 ± 0.00	0.07 ± 0.00	0.04 ± 0.00	0.03 ± 0.00	0.05 ± 0.00	0.04 ± 0.00	0.08 ± 0.01	0.06 ± 0.01	0.10 ± 0.01	0.08 ± 0.00
3191	Chlorogenic acid	4.96 ± 0.03	3.91 ± 0.03	6.27 ± 0.05	4.94 ± 0.04	2.88 ± 0.03	2.27 ± 0.04	1.48 ± 0.02	1.17 ± 0.03	4.40 ± 0.04	3.48 ± 0.04	3.79 ± 0.04	2.99 ± 0.04	2.14 ± 0.03	1.69 ± 0.05	2.40 ± 0.04	1.89 ± 0.03

nd—not detected.

**Table 2 plants-11-03219-t002:** Identified volatile compounds in peach fruit (WF) and pulp (FP) of 8 varieties analyzed by HS-SPME-GC-MS. The results are given as % of total ion current.

Name	RI_lit_	Ri_calc_	Filina		Gergana		Ufo 4		Laskava		July Lady		Flat Queen		Evmolpiya		Morsiani 90	
			WF	FP	WF	FP	WF	FP	WF	FP	WF	FP	WF	FP	WF	FP	WF	FP
**Aldehydes**																		
Pentanal	738	741	0.77 ± 0.09	0.69 ± 0.09	1.22 ± 0.15	1.10 ± 0.14	0.28 ± 0.03	0.25 ± 0.03	1.25 ± 0.15	1.13 ± 0.14	1.02 ± 0.13	0.92 ± 0.11	1.70 ± 0.21	1.53 ± 0.19	0.88 ± 0.11	0.80 ± 0.1	1.09 ± 0.13	0.98 ± 0.12
Hexanal	800	798	2.14 ± 0.26	1.93 ± 0.24	5.88 ± 0.73	5.29 ± 0.65	3.51 ± 0.43	3.16 ± 0.39	4.85 ± 0.6	4.36 ± 0.54	6.84 ± 0.84	6.15 ± 0.76	2.95 ± 0.36	2.65 ± 0.33	2.47 ± 0.3	2.22 ± 0.27	5.13 ± 0.63	4.61 ± 0.57
(E)-2-Hexenal	849	850	3.12 ± 0.38	2.81 ± 0.35	4.20 ± 0.52	3.78 ± 0.47	1.48 ± 0.18	1.34 ± 0.16	6.10 ± 0.75	5.49 ± 0.68	2.25 ± 0.28	2.02 ± 0.25	4.83 ± 0.6	4.35 ± 0.54	3.59 ± 0.44	3.23 ± 0.4	5.04 ± 0.62	4.54 ± 0.56
Heptanal	907	909	4.38 ± 0.54	3.94 ± 0.49	1.55 ± 0.19	1.39 ± 0.17	1.74 ± 0.21	1.57 ± 0.19	1.26 ± 0.16	1.13 ± 0.14	1.50 ± 0.19	1.35 ± 0.17	2.35 ± 0.29	2.11 ± 0.26	3.30 ± 0.41	2.97 ± 0.37	1.38 ± 0.17	1.24 ± 0.15
Benzaldehyde *	948	946	0.79 ± 0.1	0.71 ± 0.09	0.51 ± 0.06	0.46 ± 0.06	0.58 ± 0.07	0.52 ± 0.06	0.53 ± 0.07	0.47 ± 0.06	0.45 ± 0.06	0.41 ± 0.05	0.71 ± 0.09	0.64 ± 0.08	0.90 ± 0.11	0.81 ± 0.1	0.46 ± 0.06	0.41 ± 0.05
(E)-2-Heptenal	960	960	0.56 ± 0.07	0.50 ± 0.06	0.36 ± 0.05	0.33 ± 0.04	1.80 ± 0.22	1.62 ± 0.2	1.64 ± 0.2	1.47 ± 0.18	1.57 ± 0.19	1.41 ± 0.17	0.51 ± 0.06	0.46 ± 0.06	0.64 ± 0.08	0.58 ± 0.07	1.42 ± 0.18	1.28 ± 0.16
Octanal	999	1000	1.17 ± 0.14	1.05 ± 0.13	0.77 ± 0.09	0.69 ± 0.09	0.86 ± 0.11	0.78 ± 0.1	0.79 ± 0.1	0.71 ± 0.09	0.75 ± 0.09	0.68 ± 0.08	1.06 ± 0.13	0.96 ± 0.12	1.35 ± 0.17	1.21 ± 0.15	0.68 ± 0.08	0.61 ± 0.08
(E)-2-Octenal	1051	1047	1.57 ± 0.19	1.42 ± 0.17	1.03 ± 0.13	0.93 ± 0.11	0.64 ± 0.08	0.58 ± 0.07	3.59 ± 0.44	3.23 ± 0.4	2.43 ± 0.3	2.19 ± 0.27	1.43 ± 0.18	1.29 ± 0.16	1.81 ± 0.22	1.63 ± 0.2	3.12 ± 0.38	2.81 ± 0.35
2-methyl-Benzaldehyde	1070	1073	0.92 ± 0.11	0.83 ± 0.1	0.60 ± 0.07	0.54 ± 0.07	0.59 ± 0.07	0.53 ± 0.07	1.88 ± 0.23	1.69 ± 0.21	0.51 ± 0.06	0.46 ± 0.06	0.83 ± 0.1	0.75 ± 0.09	1.06 ± 0.13	0.95 ± 0.12	1.64 ± 0.2	1.47 ± 0.18
4-methyl-Benzaldehyde *	1084	1085	0.26 ± 0.03	0.23 ± 0.03	0.17 ± 0.02	0.15 ± 0.02	1.58 ± 0.2	1.42 ± 0.18	1.44 ± 0,18	1.29 ± 0.16	1.38 ± 0.17	1.24 ± 0.15	0.24 ± 0.03	0.21 ± 0.03	0.30 ± 0.04	0.27 ± 0.03	1.25 ± 0.15	1.13 ± 0.14
Nonanal	1102	1104	4.28 ± 0.53	3.85 ± 0.48	2.32 ± 0.29	2.09 ± 0.26	2.61 ± 0.32	2.35 ± 0.29	2.19 ± 0.27	1.97 ± 0.24	2.00 ± 0.25	1.80 ± 0.22	1.59 ± 0.2	1.43 ± 0.18	1.62 ± 0,2	1.46 ± 0.18	2.06 ± 0.25	1.86 ± 0.23
(E)-2-Nonenal	1160	1159	1.72 ± 0.21	1.55 ± 0.19	1.13 ± 0.14	1.02 ± 0.13	2.66 ± 0.33	2.40 ± 0.3	2.42 ± 0.3	2.18 ± 0.27	2.13 ± 0.26	1.92 ± 0.24	1.36 ± 0.17	1.22 ± 0.15	1.98 ± 0.24	1.78 ± 0.22	2.10 ± 0.26	1.89 ± 0.23
Decanal	1204	1205	0.42 ± 0.05	0.38 ± 0.05	0.28 ± 0.03	0.25 ± 0.03	0.31 ± 0.04	0.28 ± 0.03	0.28 ± 0.04	0.26 ± 0.03	0.27 ± 0.03	0.25 ± 0.03	0.39 ± 0.05	0.35 ± 0.04	0.49 ± 0.06	0.44 ± 0.05	0.25 ± 0.03	0.22 ± 0.03
(E)-2-Decenal	1250	1253	0.35 ± 0.04	0.53 ± 0.07	0.23 ± 0.03	0.20 ± 0.03	0.65 ± 0.08	0.58 ± 0.07	1.50 ± 0.18	1.35 ± 0.17	1.43 ± 0.18	1.29 ± 0.16	0.31 ± 0.04	0.28 ± 0.03	0.40 ± 0.05	0.36 ± 0.04	1.30 ± 0.16	1.17 ± 0.14
**Ketones**																		
3-Octanone	975	977	0.78 ± 0.1	0.86 ± 0.11	0.51 ± 0.06	0.56 ± 0.07	0.58 ± 0.07	0.63 ± 0.08	0.27 ± 0.03	0.30 ± 0.04	0.37 ± 0.05	0.40 ± 0.05	0.71 ± 0.09	0.78 ± 0.1	0.90 ± 0.11	0.99 ± 0.12	0.45 ± 0.06	0.50 ± 0.06
2-Octanone	991	992	0.61 ± 0.08	0.67 ± 0.08	0.40 ± 0.05	0.44 ± 0.05	0.45 ± 0.06	0.50 ± 0.06	0.15 ± 0.02	0.17 ± 0.02	0.39 ± 0.05	0.43 ± 0.05	0.56 ± 0.07	0.61 ± 0.08	0.70 ± 0.09	0.77 ± 0.1	0.36 ± 0.04	0.39 ± 0.05
γ-hexalactone	1045	1045	0.32 ± 0.04	0.35 ± 0.04	0.21 ± 0.03	0.23 ± 0.03	0.24 ± 0.03	0.26 ± 0.03	0.46 ± 0.06	0.51 ± 0.06	0.12 ± 0.01	0.13 ± 0.02	0.29 ± 0.04	0.32 ± 0.04	0.37 ± 0.05	0.40 ± 0.05	0.19 ± 0.02	0.20 ± 0.03
2-Nonanone *	1090	1088	0.70 ± 0.09	0.77 ± 0.1	0.46 ± 0.06	0.51 ± 0.06	0.52 ± 0.06	0.57 ± 0.07	0.16 ± 0.02	0.17 ± 0.02	0.40 ± 0.05	0.44 ± 0.05	0.64 ± 0.08	0.70 ± 0.09	0.81 ± 0.1	0.89 ± 0.11	0.41 ± 0.05	0.45 ± 0.06
γ-octalactone	1250	1251	1.95 ± 0.24	2.15 ± 0.27	1.28 ± 0.16	1.41 ± 0.17	0.63 ± 0.08	0.70 ± 0.09	3.08 ± 0.38	3.39 ± 0.42	2.46 ± 0.3	2.71 ± 0.33	1.78 ± 0.22	1.95 ± 0.24	1.15 ± 0.14	1.26 ± 0.16	2.24 ± 0.28	2.46 ± 0.3
γ-decalactone	1461	1464	1.22 ± 0.15	1.34 ± 0.17	1.59 ± 0.2	1.75 ± 0.22	1.79 ± 0.22	1.97 ± 0.24	1.63 ± 0.2	1.79 ± 0.22	1.26 ± 0.16	1.38 ± 0.17	1.11 ± 0.14	1.22 ± 0.15	1.40 ± 0.17	1.54 ± 0.19	1.42 ± 0.17	1.56 ± 0.19
γ-dodecalactone	1673	1675	1.62 ± 0.2	1.78 ± 0.22	2.65 ± 0.33	2.91 ± 0.36	4.40 ± 0.54	4.84 ± 0.6	2.00 ± 0.25	2.20 ± 0.27	2.88 ± 0.36	3.17 ± 0.39	1.47 ± 0.18	1.62 ± 0.2	1.86 ± 0.23	2.05 ± 0.25	3.48 ± 0.43	3.83 ± 0.47
**Alcohols**																		
Pentanol	770	772	0.98 ± 0.12	1.08 ± 0.13	1.18 ± 0.15	1.29 ± 0.16	1.33 ± 0.16	1.46 ± 0.18	1.21 ± 0.15	1.33 ± 0.16	1.11 ± 0.14	1.22 ± 0.15	1.63 ± 0.2	1.80 ± 0.22	2.51 ± 0.31	2.76 ± 0.34	1.05 ± 0.13	1.15 ± 0.14
Hexanol *	851	848	0.30 ± 0.04	0.33 ± 0.04	0.36 ± 0.04	0.39 ± 0.05	0.40 ± 0.05	0.44 ± 0.05	0.15 ± 0.02	0.16 ± 0.02	0.35 ± 0.04	0.38 ± 0.05	0.49 ± 0.06	0.54 ± 0.07	0.63 ± 0.08	0.69 ± 0.08	0.32 ± 0.04	0.35 ± 0.04
Heptanol	920	921	0.44 ± 0.05	0.49 ± 0.06	0.53 ± 0.07	0.59 ± 0.07	0.60 ± 0.07	0.66 ± 0.08	0.55 ± 0.07	0.60 ± 0.07	0.52 ± 0.06	0.57 ± 0.07	0.74 ± 0.09	0.81 ± 0.1	0.94 ± 0.12	1.03 ± 0.13	0.47 ± 0.06	0.52 ± 0.06
Benzyl Alchool *	1035	1035	0.15 ± 0.02	0.17 ± 0.02	0.19 ± 0.02	0.20 ± 0.03	1.13 ± 0.14	1.25 ± 0.15	1.03 ± 0.13	1.13 ± 0.14	0.99 ± 0.12	1.08 ± 0.13	0.26 ± 0.03	0.28 ± 0.03	0.32 ± 0.04	0.36 ± 0.04	0.90 ± 0.11	0.99 ± 0.12
Nonanol	1149	1150	0.82 ± 0.1	0.90 ± 0.11	0.98 ± 0.12	1.08 ± 0.13	1.10 ± 0.14	1.22 ± 0.15	0.43 ± 0.05	0.47 ± 0.06	0.96 ± 0.12	1.06 ± 0.13	1.36 ± 0.17	1.50 ± 0.18	1.72 ± 0.21	1.89 ± 0.23	0.87 ± 0.11	0.96 ± 0.12
**Acids**																		
Butanoic acid	759	760	1.59 ± 0.2	1.75 ± 0.22	1.43 ± 0.18	1.58 ± 0.19	2.64 ± 0.33	2.90 ± 0.36	3.31 ± 0.41	3.64 ± 0.45	3.17 ± 0.39	3.48 ± 0.43	1.99 ± 0.25	2.19 ± 0.27	2.52 ± 0.31	2.77 ± 0.34	2.88 ± 0.36	3.17 ± 0.39
2-methyl-Pentanoic acid	926	924	1.41 ± 0.17	1.56 ± 0.19	2.86 ± 0.35	3.14 ± 0.39	2.22 ± 0.27	2.44 ± 0.3	2.93 ± 0.36	3.22 ± 0.4	2.80 ± 0.35	3.08 ± 0.38	1.77 ± 0.22	1.95 ± 0.24	2.24 ± 0.28	2.46 ± 0.30	2.55 ± 0.31	2.80 ± 0.35
Hexanoic acid	964	966	2.08 ± 0.26	2.28 ± 0.28	4.04 ± 0.5	4.44 ± 0.55	5.53 ± 0.68	6.08 ± 0.75	1.34 ± 0.17	1.48 ± 0.18	6.54 ± 0.81	7.20 ± 0.89	2.59 ± 0.32	2.85 ± 0.35	3.28 ± 0.41	3.61 ± 0.45	3.95 ± 0.49	4.34 ± 0.54
Octanoic acid	1165	1166	1.39 ± 0.17	1.53 ± 0.19	1.25 ± 0.15	1.38 ± 0.17	2.80 ± 0.35	3.08 ± 0.38	2.55 ± 0.31	2.80 ± 0.16	2.44 ± 0.3	2.68 ± 0.33	1.74 ± 0.21	1.91 ± 0.24	2.20 ± 0.27	2.42 ± 0.3	2.21 ± 0.27	2.44 ± 0.3
Nonanoic acid	1270	1272	2.39 ± 0.29	2.62 ± 0.32	2.15 ± 0.27	2.36 ± 0.29	1.42 ± 0.18	1.56 ± 0.19	1.32 ± 0.16	1.45 ± 0.18	2.10 ± 0.26	2.31 ± 0.29	2.98 ± 0.37	3.28 ± 0.4	3.77 ± 0.47	4.15 ± 0.51	1.91 ± 0.24	2.10 ± 0.26
*n*-Decanoic acid	1367	1368	2.03 ± 0.25	2.23 ± 0.28	1.83 ± 0.23	2.01 ± 0.25	1.06 ± 0.13	1.16 ± 0.14	1.18 ± 0.15	1.29 ± 0.16	1.79 ± 0.22	1.97 ± 0.24	2.53 ± 0.31	2.79 ± 0.34	3.21 ± 0.4	3.53 ± 0.44	1.63 ± 0.2	1.79 ± 0.22
Dodecanoic acid	1558	1559	2.60 ± 0.32	2.86 ± 0.35	2.34 ± 0.29	2.57 ± 0.32	1.64 ± 0.2	1.80 ± 0.22	1.30 ± 0.16	1.43 ± 0.18	2.29 ± 0.28	2.52 ± 0.31	3.25 ± 0.4	3.57 ± 0.44	2.11 ± 0.26	2.32 ± 0.29	2.08 ± 0.26	2.29 ± 0.28
*n*-Hexadecanoic acid *	1960	1960	1.04 ± 0.13	1.14 ± 0.14	0.93 ± 0.12	1.03 ± 0.13	0.61 ± 0.07	0.67 ± 0.08	0.54 ± 0.07	0.59 ± 0.07	0.91 ± 0.11	1.01 ± 0.12	1.30 ± 0.16	1.43 ± 0.18	1.64 ± 0.2	1.80 ± 0.22	0.83 ± 0.1	0.91 ± 0.11
**Esters**																		
Ethyl acetate *	607	610	1.76 ± 0.22	1.59 ± 0.2	1.95 ± 0.24	1.75 ± 0.22	2.19 ± 0.27	1.97 ± 0.24	1.99 ± 0.25	1.80 ± 0.22	1.91 ± 0.24	1.72 ± 0.21	1.60 ± 0.2	1.44 ± 0.18	2.03 ± 0.25	1.83 ± 0.23	1.73 ± 0.21	1.56 ± 0.19
Ethyl pentanoate	903	905	1.42 ± 0.17	1.27 ± 0.16	1.72 ± 0.21	1.55 ± 0.19	1.94 ± 0.24	1.74 ± 0.22	1.76 ± 0.22	1.58 ± 0.2	1.68 ± 0.21	1.52 ± 0.19	1.29 ± 0.16	1.16 ± 0.14	1.63 ± 0.2	1.47 ± 0.18	1.53 ± 0.19	1.38 ± 0.17
Ethyl tiglate	940	938	5.02 ± 0.62	4.52 ± 0.56	3.43 ± 0.42	3.09 ± 0.38	3.86 ± 0.48	3.48 ± 0.43	3.51 ± 0.43	3.16 ± 0.39	3.36 ± 0.41	3.02 ± 0.37	2.76 ± 0.34	2.49 ± 0.31	4.02 ± 0.5	3.62 ± 0.45	3.06 ± 0.38	2.75 ± 0.34
Ethyl hexanoate	998	886	4.39 ± 0.54	3.95 ± 0.49	5.05 ± 0.62	4.54 ± 0.56	1.75 ± 0.22	1.58 ± 0.19	3.19 ± 0.39	2.87 ± 0.35	1.52 ± 0.19	1.37 ± 0.17	1.99 ± 0.25	1.79 ± 0.22	2.05 ± 0.25	1.85 ± 0.23	4.14 ± 0.51	3.72 ± 0.46
Ethyl Heptanoate	1096	1097	2.29 ± 0.28	2.06 ± 0.25	1.50 ± 0.19	1.35 ± 0.17	1.69 ± 0.21	1.52 ± 0.19	1.54 ± 0.19	1.38 ± 0.17	1.47 ± 0.18	1.32 ± 0.16	2.08 ± 0.26	1.88 ± 0.23	2.64 ± 0.33	2.37 ± 0.29	1.34 ± 0.16	1.20 ± 0.15
Ethyl Benzoate	1170	1173	3.94 ± 0.49	3.54 ± 0.44	2.58 ± 0.32	2.32 ± 0.29	2.90 ± 0.36	2.61 ± 0.32	1.63 ± 0.2	1.47 ± 0.18	2.52 ± 0.31	2.27 ± 0.28	3.58 ± 0.44	3.22 ± 0.4	1.23 ± 0.15	1.10 ± 0.14	2.30 ± 0.28	2.07 ± 0.26
Ethyl Octanoate	1195	1198	2.25 ± 0.28	2.02 ± 0.25	2.19 ± 0.27	1.97 ± 0.24	2.47 ± 0.3	2.22 ± 0.27	2.03 ± 0.25	1.82 ± 0.23	1.81 ± 0.22	1.63 ± 0.2	3.04 ± 0.38	2.74 ± 0.34	2.58 ± 0.32	2.32 ± 0.29	1.95 ± 0.24	1.76 ± 0.22
Methyl Nonanoate *	1226	1225	1.87 ± 0.23	1.68 ± 0.21	2.81 ± 0.35	2.53 ± 0.31	0.96 ± 0.12	0.87 ± 0.11	2.88 ± 0.36	2.59 ± 0.32	2.75 ± 0.34	2.48 ± 0.31	1.70 ± 0.21	1.53 ± 0.19	2.15 ± 0.27	1.93 ± 0.24	1.50 ± 0.19	1.35 ± 0.17
Ethyl oct-(2E)-enoate	1242	1240	1.71 ± 0.21	1.54 ± 0.19	1.12 ± 0.14	1.01 ± 0.12	1.26 ± 0.16	1.14 ± 0.14	1.15 ± 0.14	1.03 ± 0.13	1.01 ± 0.12	0.91 ± 0.11	1.55 ± 0.19	1.40 ± 0.17	0.87 ± 0.11	0.78 ± 0.1	1.00 ± 0.12	0.90 ± 0.11
1-Octen-3-yl-butanoate	1280	1280	2.41 ± 0.3	2.17 ± 0.27	1.58 ± 0.19	1.42 ± 0.18	0.68 ± 0.08	0.61 ± 0.08	1.61 ± 0.2	1.45 ± 0.18	1.45 ± 0.18	1.31 ± 0.16	2.19 ± 0.27	1.97 ± 0.24	1.67 ± 0.21	1.50 ± 0.19	1.40 ± 0.17	1.26 ± 0.16
Methyl Decanoate	1320	1322	1.28 ± 0.16	1.15 ± 0.14	0.84 ± 0.1	0.76 ± 0.09	0.79 ± 0.1	0.71 ± 0.09	0.72 ± 0.09	0.65 ± 0.08	0.69 ± 0.08	0.62 ± 0.08	1.16 ± 0.14	1.05 ± 0.13	1.47 ± 0.18	1.33 ± 0.16	0.62 ± 0.08	0.56 ± 0.07
Benzyl butanoate	1344	1345	1.37 ± 0.17	1.23 ± 0.15	2.48 ± 0.31	2.23 ± 0.28	0.84 ± 0.1	0.75 ± 0.09	0.76 ± 0.09	0.68 ± 0.08	0.73 ± 0.09	0.65 ± 0.08	1.24 ± 0.15	1.12 ± 0.14	0.47 ± 0.06	0.43 ± 0.05	0.66 ± 0.08	0.59 ± 0.07
(2E)-Octenyl butanoate	1388	1385	1.79 ± 0.22	1.61 ± 0.2	1.17 ± 0.14	1.06 ± 0.13	1.32 ± 0.16	1.19 ± 0.15	1.20 ± 0.15	1.08 ± 0.13	1.15 ± 0.14	1.04 ± 0.13	1.63 ± 0.2	1.47 ± 0.18	2.06 ± 0.25	1.86 ± 0.23	1.05 ± 0.13	0.94 ± 0.12
Linalool butanoate	1423	1425	1.31 ± 0.16	1.17 ± 0.15	2.30 ± 0.28	2.07 ± 0.26	1.49 ± 0.18	1.34 ± 0.17	2.35 ± 0.29	2.12 ± 0.26	2.12 ± 0.26	1.91 ± 0.24	3.19 ± 0.39	2.87 ± 0.35	1.50 ± 0.19	1.35 ± 0.17	2.04 ± 0.25	1.84 ± 0.23
2-Phenyl ethyl butanoate	1435	1436	3.02 ± 0.37	2.72 ± 0.34	1.98 ± 0.24	1.78 ± 0.22	2.23 ± 0.28	2.01 ± 0.25	2.03 ± 0.25	1.82 ± 0.23	1.94 ± 0.24	1.74 ± 0.22	2.75 ± 0.34	2.47 ± 0.31	3.47 ± 0.43	3.13 ± 0.39	1.76 ± 0.22	1.59 ± 0.2
2-Phenyl propyl butanoate	1482	1480	2.05 ± 0.25	1.84 ± 0.23	1.34 ± 0.17	1.21 ± 0.15	2.71 ± 0.33	2.44 ± 0.3	1.37 ± 0.17	1.23 ± 0.15	1.31 ± 0.16	1.18 ± 0.15	1.86 ± 0.23	1.67 ± 0.21	2.35 ± 0.29	2.12 ± 0.26	1.19 ± 0.15	1.07 ± 0.13
**Hydrocarbones**																		
Undecane *	1098	1095	1.17 ± 0.14	1.29 ± 0.16	0.77 ± 0.09	0.84 ± 0.1	0.86 ± 0.11	0.95 ± 0.12	0.58 ± 0.07	0.64 ± 0.08	0.48 ± 0.06	0.52 ± 0.06	1.06 ± 0.13	1.17 ± 0.14	0.25 ± 0.03	0.27 ± 0.03	0.68 ± 0.08	0.75 ± 0.09
Dodecane *	1200	1202	1.21 ± 0.15	1.33 ± 0.16	1.37 ± 0.17	1.51 ± 0.19	1.54 ± 0.19	1.70 ± 0.21	1.40 ± 0.17	1.87 ± 0.23	1.30 ± 0.16	1.43 ± 0.18	1.51 ± 0.19	1.66 ± 0.2	0.81 ± 0.1	0.89 ± 0.11	1.22 ± 0.15	1.34 ± 0.17
Tridecane *	1302	1304	0.36 ± 0.05	0.54 ± 0.07	1.82 ± 0.23	2.01 ± 0.25	2.05 ± 0.25	2.26 ± 0.28	0.77 ± 0.09	0.84 ± 0.1	1.58 ± 0.19	1.74 ± 0.21	0.33 ± 0.04	0.36 ± 0.05	0.42 ± 0.05	0.46 ± 0.06	1.62 ± 0.2	1.79 ± 0.22
Tetradecane *	1400	1401	1.72 ± 0.21	1.90 ± 0.23	2.43 ± 0.3	2.67 ± 0.33	2.25 ± 0.28	2.48 ± 0.31	1.39 ± 0.17	1.65 ± 0.2	1.04 ± 0.13	1.14 ± 0.14	2.03 ± 0.25	2.24 ± 0.28	1.14 ± 0.14	1.25 ± 0.15	2.16 ± 0.27	2.38 ± 0.29
Pentadecane *	1497	1495	1.06 ± 0.13	1.22 ± 0.15	0.70 ± 0.09	0.77 ± 0.09	0.78 ± 0.1	0.86 ± 0.11	0.71 ± 0.09	0.88 ± 0.11	0.68 ± 0.08	0.75 ± 0.09	0.97 ± 0.12	1.06 ± 0.13	0.67 ± 0.08	0.74 ± 0.09	0.62 ± 0.08	0.68 ± 0.08
Hexadecane *	1600	1601	0.58 ± 0.07	0.63 ± 0.08	0.38 ± 0.05	0.41 ± 0.05	0.42 ± 0.05	0.47 ± 0.06	0.39 ± 0.05	0.44 ± 0.05	0.24 ± 0.03	0.26 ± 0.03	0.52 ± 0.06	0.58 ± 0.07	0.66 ± 0.08	0.73 ± 0.09	0.34 ± 0.04	0.37 ± 0.05
Heptadecane *	1701	1700	0.93 ± 0.12	1.10 ± 0.14	0.61 ± 0.08	0.67 ± 0.08	0.69 ± 0.08	0.76 ± 0.09	0.28 ± 0.03	0.35 ± 0.04	0.46 ± 0.06	0.51 ± 0.06	0.85 ± 0.1	0.93 ± 0.12	1.07 ± 0.13	1.18 ± 0.15	0.54 ± 0.07	0.60 ± 0.07
**Terpenes**																		
β-Myrcene *	980	985	1.24 ± 0.15	1.46 ± 0.18	2.60 ± 0.32	2.92 ± 0.36	1.00 ± 0.12	1.10 ± 0.14	2.36 ± 0.29	2.60 ± 0.32	0.77 ± 0.09	0.87 ± 0.11	1.61 ± 0.2	1.77 ± 0.22	0.44 ± 0.05	0.47 ± 0.06	1.36 ± 0.17	1.50 ± 0.19
p-Cymene	1018	1020	0.17 ± 0.02	0.19 ± 0.02	1.10 ± 0.14	1.35 ± 0.17	1.25 ± 0.15	1.38 ± 0.17	0.40 ± 0.05	0.44 ± 0.05	0.70 ± 0.09	0.77 ± 0.1	1.75 ± 0.22	1.93 ± 0.24	0.20 ± 0.02	0.21 ± 0.03	0.99 ± 0.12	1.09 ± 0.13
Limonene *	1024	1022	0.72 ± 0.09	0.80 ± 0.1	1.50 ± 0.19	1.65 ± 0.2	3.58 ± 0.44	3.94 ± 0.49	1.64 ± 0.2	1.80 ± 0.22	2.21 ± 0.27	2.43 ± 0.3	0.66 ± 0.08	0.72 ± 0.09	0.86 ± 0.11	0.95 ± 0.12	1.83 ± 0.23	2.02 ± 0.25
(Z)-β-Ocimene	1035	1036	0.87 ± 0.11	0.96 ± 0.12	0.57 ± 0.07	0.63 ± 0.08	0.64 ± 0.08	0.71 ± 0.09	0.28 ± 0.03	0.31 ± 0.04	0.56 ± 0.07	0.61 ± 0.08	0.79 ± 0.1	0.88 ± 0.11	1.00 ± 0.12	1.10 ± 0.14	0.51 ± 0.06	0.56 ± 0.07
(E)-β-Ocimene *	1042	1041	1.56 ± 0.19	1.71 ± 0.21	1.02 ± 0.13	1.12 ± 0.14	1.15 ± 0.14	1.26 ± 0.16	1.00 ± 0.12	1.09 ± 0.14	0.74 ± 0.09	0.82 ± 0.1	1.42 ± 0.17	1.56 ± 0.19	1.82 ± 0.22	2.00 ± 0.25	0.91 ± 0.11	1.00 ± 0.12
Linalol	1093	1094	2.44 ± 0.3	2.74 ± 0.34	1.95 ± 0.24	2.15 ± 0.27	2.20 ± 0.27	2.42 ± 0.3	1.63 ± 0.2	1.79 ± 0.22	1.79 ± 0.22	1.97 ± 0.24	1.22 ± 0.15	1.34 ± 0.17	2.03 ± 0.25	2.23 ± 0.28	1.47 ± 0.18	1.62 ± 0.2
(Z)-β-Farnesene	1440	1443	0.29 ± 0.04	0.32 ± 0.04	0.91 ± 0.11	1.00 ± 0.12	1.03 ± 0.13	1.13 ± 0.14	0.89 ± 0.11	0.97 ± 0.12	0.79 ± 0.1	0.87 ± 0.11	1.26 ± 0.16	1.39 ± 0.17	0.34 ± 0.04	0.37 ± 0.05	0.81 ± 0.1	0.89 ± 0.11
(E)-β-Farnesene	1452	1455	0.64 ± 0.08	0.87 ± 0.11	0.43 ± 0.05	0.48 ± 0.06	0.27 ± 0.03	0.30 ± 0.04	0.90 ± 0.11	1.02 ± 0.13	0.54 ± 0.07	0.60 ± 0.07	0.60 ± 0.07	0.66 ± 0.08	0.76 ± 0.09	0.84 ± 0.1	0.39 ± 0.05	0.42 ± 0.05

* compounds identified by checking with standards.

## Data Availability

The data presented in this study are available on request from the corresponding author.

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
