# Peer review of "Pattern Recognition of Varieties of Peach Fruit and Pulp from Their Volatile Components and Metabolic Profile Using HS-SPME-GC/MS Combined with Multivariable Statistical Analysis"

_plants, 2022, doi:10.3390/plants11233219_

Round 1
Reviewer 1 Report
The manuscript “Pattern recognition of varieties of peach fruit and pulp from their volatile components using HS-SPME-GC/MS combined with multivariable statistical analys" is devoted to a study of volatiles from eight peach cultivars. Compounds were extracted from both peach fruit and pulp. GC-MS analysis was used to identify all compounds. Principal component analysis and hierarchical cluster analysis were performed to analyze obtained data. Unfortunately, in general, the work does not seem to be very large in terms of the amount and significance of the data obtained. The importance of the conclusions drawn is rather narrow. It may be necessary to expand this study and consider, for example, comparison by other parameters (the amount of biologically active substances, sugars, vitamins).
I think, this manuscript can be published in the Plants after major revision taking into account general recommendation and some of the remarks described below:
1. Section Fruit Samples: at what exact time (month) were the samples collected and why was this particular collection time chosen?
2. How many repetitions of the analysis were done? It is required to specify the standard deviation in Table 1.
3. The Title of the manuscript: “analysis”, not “analys”.
4. The text in the figures should be enlarged. In this display, it is poorly readable.
Author Response
The authors would like to show their appreciation for the reviewer’s helpful, attentive, and thorough review. We are delighted that we have provided an interesting scientific topic. We accept his\her very detailed suggestions and improvements for the manuscript.
Below, the reviewer can find a point by point answer to all comments:
In general, the work does not seem to be very large in terms of the amount and significance of the data obtained. The importance of the conclusions drawn is rather narrow. It may be necessary to expand this study and consider, for example, comparison by other parameters (the amount of biologically active substances, sugars, vitamins). - Information about the sugars, vitamins, and biological activities of the same peach varieties have been discussed in some previously published works i.e. 10.3390/foods10010164; 10.3390/molecules26092818; 10.3390/molecules26144183. This is the reason why we narrow the amount of work, and target the distinguishment of the pulp from the whole fruit, as peels are often seen as food waste, or not consumed due to fuzz allergies, and removed prior to consumption.
Section Fruit Samples: at what exact time (month) were the samples collected and why was this particular collection time chosen? - The samples, object of analysis, were chosen throughout the summer season, aiming at having representatives from different parts of the summer season, starting with the very early, end of June (“Filina”) and ending with the late, mid-September (“Morsiani 90”). Authors were aiming at a longer time frame (June-September) and not previously chosen cultivars. The proposed cultivars represented well the peach palette and were recommended by an agronomy PhD (acknowledged in the manuscript) with long experience in the field of peach plantations.
- How many repetitions of the analysis were done? It is required to specify the standard deviation in Table 1. Table 1 has been updated with SD values (3 repetitions).
The Title of the manuscript: “analysis”, not “analys”. - A change in the title, due to a typo mistake, has been made.
The text in the figures should be enlarged. In this display, it is poorly readable. - The authors completely understand the reviewers idea about the figures. Unfortunately, Figures 2-4 are plotted directly from the Metaboanalyst software and the font cannot be adjusted. We have made them a bit bigger and hope that they are a little more readable since they hold a lot of information in them.
We very much hope that the improved manuscript will be accepted for publication.
Reviewer 2 Report
General remarks
The manuscript describes the GC-MS-based protocol to determine the olfactory properties of different peach varieties. In my opinion, this manuscript does not present any novelty and represents not up to date approach for multivariate analysis. However, I believe that after major modification it will be possible to publish this manuscript.
Below please find a detailed review.
Abstract:
‘The fruit aroma profile or ‘fruits aroma profile’.
Results and Discussion
Line 83: More volatiles – in quantitative or qualitative aspect?
Line 87: It is hard to state while you've mixed pulp and peel, and analyze volatiles only. Olfactory should also be involved in this experiment.
Table 1: the comparison of relative concentration between WF and FP is not a good way to establish aroma key compounds. Still, it is impossible to tell whether the concentration of compound A is higher in WF or FP. And this is crucial, not relative distribution.
Line 101: these might be unrelated to microbial activity
Table 1: It looks like both Figure 1 and Table 2 consist of the same information. Please decide which looks best for you. And Table 2 should be corrected (line for WF/FP line)
Line 122 (as well as in similar places): Please state that that volatiles was in headspace – it does not mean that it can be easily translated to the bulk of the fruit composition.
Line 134: Since ripening was not determined it is rather insufficient to state that it is related to volatiles
Line 165: odour threshold is not based on the relative concentration
Line 173: PCA does not produce clusters. This is a pattern recognition method.
Line 174: In Figure 2 is HCA, not PCA
Line 175: I have a doubt. What were the input data? Do those values were normalized? What are those groups mean- e.g. why Evmolpiya is an outlier in Figure A? Why July Lady and Morsiani 90 are similar in both cases?
Line 181: this stand the opposite of HCA
Figure 3: where are the replicates on the PCA biplot? It is hard to determine distribution within one variety.
Line 189: the rule of thumb suggests that at least 80% of variation should be covered.
Line 191: I don't get the relation
Figure 4: Heatmap covers the information of HCA (Figure 2). Please consider taking Figure 4 and removing Figure 2 since Figure 4 is more informative. The discussion of Figure 4 is missing - that might be the most important part of your work.
Materials and Methods:
Line 199: How many replicates were studied? How to assess the ripeness of fruit. Did you analyze the volatile composition of at least one of the peach varieties at different ripening stages? Is it possible that those changes are related to fruit growth, not variety?
Line 207: please at least describe the main parameters of SPME. How the sample was prepared?
Line 218: ob-tained
Line 221: at least, some should be checked by standards
Line 233: more data are needed- data matrix preparation, normalization procedure, and hca parameters missing here.
Conclusions
Line 233: it was already mentioned in the abstract.
Author Response
The authors would like to show their appreciation for the reviewer’s helpful, attentive, and thorough review. We are delighted that we have provided an interesting scientific topic. We accept his\her very detailed suggestions and improvements for the manuscript.
Below, the reviewer can find a point by point answer to all comments:
‘The fruit aroma profile or ‘fruits aroma profile’. In the manuscript, it is ‘The fruit’s aroma profile” (in general, not specifying) as a possessive pronoun, not a noun; that is why this is not a mistake, it was intended to be written in this way.
Line 83: More volatiles – in quantitative or qualitative aspect? Quantification should be based on calibration curves using standards for each compound. Such approach has not been pursuit in this research.
Line 87: It is hard to state while you've mixed pulp and peel, and analyze volatiles only. Olfactory should also be involved in this experiment. The pulp and peel corresponds to the whole fruit. Peels are often removed prior to consumption (for different reasons) which leaves the consumer with a different volatile experience. This is the reason why the authors have pursuit such an approach of looking at the pulp only and the whole fruit, meaning pulp and peel present together. Olfactory could not be included in this experiment since we do not support GC-O at our laboratories. This limits the study, but unfortunately the authors are always restricted to the apparatus they are provided with. We will look into the possibility for applying the proposed technique in future research topics. Hope that the authors have understood the reviewer’s comment in full.
Table 1: the comparison of relative concentration between WF and FP is not a good way to establish aroma key compounds. Still, it is impossible to tell whether the concentration of compound A is higher in WF or FP. And this is crucial, not relative distribution. Yes, this is not the best possible way to establish key components, but we still believe it can be ruled as acceptable or primary for further analysis and possible comparison. Some techniques like the evaluation of the percentage peak are for quantitative analysis are applicable, but standards are a must.
Line 101: these might be unrelated to microbial activity Yes, the reviewer is correct, that they might not contribute, but we would like to mention that possibility following the reference, which may lead a path for future research.
Table 1: It looks like both Figure 1 and Table 2 consist of the same information. Please decide which looks best for you. And Table 2 should be corrected (line for WF/FP line). The authors will remove Table 2, and keep Figure 1.
Line 122 (as well as in similar places): Please state that that volatiles was in headspace – it does not mean that it can be easily translated to the bulk of the fruit composition. The authors have made effort to clear it out, where applicable.
Line 134: Since ripening was not determined it is rather insufficient to state that it is related to volatiles. We believe that the fruits were appropriately ripened, as explained in a further comment concerning L199. However, samples at different ripening stages have not been collected for analysis.
Line 165: odour threshold is not based on the relative concentration. Yes, the reviewer is correct. The odor threshold represents the minimum concentration of a substance at which a majority of test subjects can detect and identify the characteristic odor of a substance. We use this as a hypothesis, and we have adjusted the sentence in L164-166 so that is not misleading.
Line 173: PCA does not produce clusters. This is a pattern recognition method. A correction in L173 has been made.
Line 174: In Figure 2 is HCA, not PCA. As suggested below, Figure 2 will be removed.
Line 175: I have a doubt. What were the input data? Do those values were normalized? What are those groups mean- e.g. why Evmolpiya is an outlier in Figure A? Why July Lady and Morsiani 90 are similar in both cases? Authors have used the Metaboanalyst software for data analysis, plotting 3 repetitions of the data, normalizing by log10, and auto-scaling the data.
Line 181: this stand the opposite of HCA. Hierarchical clustering analysis groups similar objects into clusters, which results in a set of clusters, where each cluster is distinct from each other cluster, and the objects within each cluster are broadly similar to each other. We believe that we have followed this in our comment.
Figure 3: where are the replicates on the PCA biplot? It is hard to determine distribution within one variety. The Metaboanalyst software provides an opportunity for presenting less data on the figures plotted in the program so that they can be more reader friendly. We have taken advantage of this feature.
Line 189: the rule of thumb suggests that at least 80% of variation should be covered. The authors stick to the data presented by the software.
Line 191: I don't get the relation The authors thought it is good to acknowledge that the score plot is not presented as figure, but if the reviewer finds this information irrelevant, we will remove it from the sentence.
Figure 4: Heatmap covers the information of HCA (Figure 2). Please consider taking Figure 4 and removing Figure 2 since Figure 4 is more informative. The discussion of Figure 4 is missing - that might be the most important part of your work. The authors agree with the reviewer’s proposition about removing Figure 2, and additionally discussing Figure 4 (now Figure 3).
Line 199: How many replicates were studied? How to assess the ripeness of fruit. Did you analyze the volatile composition of at least one of the peach varieties at different ripening stages? Is it possible that those changes are related to fruit growth, not variety? Peaches and nectarines were harvested in the morning between 6 am and 8 am on different harvest dates. No bactericides were applied to plantings during testing and no rain events within 24 h of harvesting have been registered. The fruits were free from major damages such as blemishes, flaws, and such. Forty peaches per variety were harvested and transported in pulp trays in an air-conditioned vehicle to the University of food technologies, where the fruit was randomly places in new trays in order to minimize the differences in fruit quality. Extra fruit was harvested in case there was decay or damage during/after the harvest. Fruit samples were collected by specially trained personnel at the Fruit Growing Institute. There researchers follow the ripeness of all fruit available in their planation. This study has focused only on fruit that were picked up at full ripeness according to the team from the Institute (they measure several parameters in order to decide when fruits should be harvested). There are papers that have looked into the possible changes at different ripeness stages but we believe this can be seen a future opportunity. Then it can be concluded if the established changes between the whole fruit and the fruit pulp only are due to variety, fruit growth, meteorological peculiarities, among others.
Line 207: please at least describe the main parameters of SPME. How the sample was prepared? For the purposes of headspace sampling, 2 cm SPME fiber assembly Divinylbenzene/Carboxen/Polydimethylsiloxane (DVB/CAR/PDMS, Supelco, Bellefonte, PA, USA) was used, according the procedure described by Uekane et al., 2017 [doi:10.1016/j.foodchem.2016.09.098]. Briefly, the SPME needle was introduced into the sample, and then the fiber was exposed to the headspace of 2.0 g of each fruit sample in a 20-mL sealed vial at 50°C for 30 min. After that, the fiber was thermally desorbed for 5 min into the GC injection port at 250°C with a 0.75 mm i.d. SPME liner. The sampling procedure was automatically performed with a G1888 Network Headspace Sampler, integrated on-line with the corresponding GC-MS system. An Agilent 7890A GC unit coupled to an Agilent 5975C MSD and a DB-5ms (30 m × 0.25 mm × 0.25 μm) column were used to analyze the volatile compounds in all investigated samples. The oven temperature program was as follows: from 40ºC (hold 1 min) to 250 °C (hold 5 min) at 2 °C /min; carrier gas: helium with flow rate: 1.2 mL/min; transfer line temperature: 270°C; ion source temperature: 200°C, EI energy: 70 eV, mass range: 50 to 550 m/z at 1.0 s/decade. The analyses were performed in three replications and the obtained values were expressed as the mean value and standard deviation. AMDIS software, version 2.64 (Automated Mass Spectral Deconvolution and Identification System, NIST, Gaithersburg, MD, USA) aided in the reading of the obtained mass spectra and the identification of the metabolites. AMDIS recorded the RIs of the compounds with a standard n-hydrocarbon calibration mixture (C8-C36, Restek, Teknokroma, Spain). For identification, the separated compounds were compared to their GC-MS spectra and Kovats retention index (RI) with reference compounds in the NIST’08 database (NIST Mass Spectral Database, PC-Version 5.0, 2008 from National Institute of Standards and Technology, Gaithersburg, MD, USA).
Line 218: ob-tained The typo mistake in line 218 has been corrected.
Line 221: at least, some should be checked by standards. Some of the compounds are indeed check with standards available at the laboratory, those are: benzaldehyde; 4-methylbenzaldehyde; n-hexadecanoic acid, ethyl acetate, methyl nonanoate, undecane, dodecane, tridecane, tetradecane, pentadecane, Hexadecane, Heptadecane, β-Myrcene, Limonene, and (E)-β-Ocimene. Table 1 has been updated and the abovementioned compounds are marked with an asterisk as well as a footnote has been added.
Line 233: more data are needed- data matrix preparation, normalization procedure, and hca parameters missing here. The additional information is available in the reference cited. The authors have decided to reference the detailed information in order to minimize similarities in papers.
Line 233: it was already mentioned in the abstract. We have intentionally made a repetition for two reasons: first, some readers get only access to the abstract of a published work, while others tend to focus on the conclusions section in order to decide wheatear to focus on the full text or not. This is why we would like to keep this repetition, as it is acceptable to sum up as much possible in this section without making it too long.
We very much hope that the improved manuscript will be reconsidered for publication.

Round 2
Reviewer 1 Report
The authors have done some work to improve the manuscript. The data presented is a more correct representation. Unfortunately, the figures copied from the Metaboanalyst software are still not readable. Perhaps you should use another visualization program.
However, despite this, the work does not appear to have much significance in its present form. The GC-MS analysis data are some continuation of previous work.
I would advise the authors to reconsider the approach to data collection and expand the article. This may require collecting samples next season. I believe that the work in its present form can still be published after a major revision or rejected.
Author Response
The authors would like to express their gratitude to the reviewer for finding the time to re-evaluate the manuscript.
Below are our answers to the proposed corrections:
The Metaboanalyst figures are presented in a larger scale making them more readable. The authors can provide them as supplementary files if the reviewer find it more suitable.
Authors have introduced new data considering the polar and lipid metabolites identified using the same methods.
We very much hope that the improved manuscript with the data added, will be seen as suitable for publication.
Reviewer 2 Report
Thank you for addressing all of my comments.
Author Response
The authors would like to express their gratitude to the reviewer for finding the time to re-evaluate the manuscript.
Round 3
Reviewer 1 Report
I thank the authors for their work.
The data presented in the manuscript are expanded and become more important. The figures have become somewhat more readable, however, I strongly recommend that the authors remove Figures 2 and 4 from the manuscript. The names of the components on them overlap each other, so there is no informativeness in them.
Taking into account the above, I suppose that the manuscript can be published after a minor revision.
Author Response
The authors would like to express their gratitude to the reviewer for his/her continued engagement in the review process.
Below the reviewer can find a point-by-point answer to his/her comments:
The data presented in the manuscript are expanded and become more important. The figures have become somewhat more readable, however, I strongly recommend that the authors remove Figures 2 and 4 from the manuscript. The names of the components on them overlap each other, so there is no informativeness in them. – Figures 2 and 4 have been removed from the manuscript.
The authors hope that the revised manuscript will be seen as suitable for publication.